# Investigating the Effectiveness of Multiple Expert Models Collaboration

**Ikumi Ito**[1]    **Takumi Ito**[1,2]    **Jun Suzuki**[1,3]    **Kentaro Inui**[4,1,3]

[1] Tohoku University    [2] Langsmith Inc.    [3] RIKEN    [4] MBZUAI

ikumi.ito.p8@dc.tohoku.ac.jp

{t-ito, jun.suzuki}@tohoku.ac.jp  kentaro.inui@mbzuai.ac.ae

## Abstract

This paper aims to investigate the effectiveness of several machine translation (MT) models and aggregation methods in a multi-domain setting under fair conditions and explore a direction for tackling multi-domain MT. We mainly compare the performance of the single model approach by jointly training all domains and the multi-expert models approach with a particular aggregation strategy. We conduct experiments on multiple domain datasets and demonstrate that a combination of smaller domain expert models can outperform a larger model trained for all domain data.

## 1 Introduction

The machine translation (MT) research community has been paying much attention to multi-domain evaluation (Saunders, 2022; Pham et al., 2021) as well as document-level evaluation these days. This trend is supported by the fact that the main translation task in the WMT competition[1] shifted to a multi-domain setting after 2022. We explore a more effective and efficient approach since this is a more realistic and desirable setting in actual MT systems.

There are two major approaches for tackling multi-domain adaptation: a Multi-Domain Model (MDM), and Domain Expert Models (DEMs). In MDM, the model is trained on all domain data; in DEMs, expert models are trained for each domain (Figure 1). Most previous studies have focused on MDM (Kobus et al., 2017; Britz et al., 2017) since MDM is the identical approach when we do not consider domains. In contrast, DEMs have yet to be investigated in-depth due to not much attention in the community until recent years. We hypothesize that DEMs have the potential to achieve superior performance if each expert model is well-trained in the corresponding domain and better combined.

[1] http://www2.statmt.org

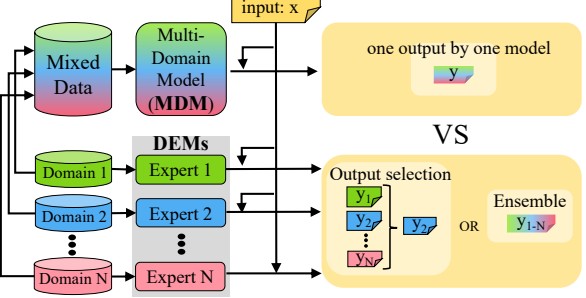

Figure 1: Comparative overview of MDM (top) and DEMs (bottom) settings.

Moreover, DEMs may have several advantages over MDMs. Thus, verifying DEMs' potential advantages would be worthwhile.

In this study, we aim to compare MDM and DEMs with various configurations in a fair condition and to reveal the effectiveness of DEMs in the current MT technologies. We conduct experiments on multiple domain datasets and demonstrate that a combination of smaller DEMs can outperform a single larger MDM.

## 2 Related Work

**Multi-domain.** The straightforward approach to addressing multiple domains is training on data encompassing various domains. Much research is underway to develop a multi-domain model, such as adding domain-specific tags to inputs (Kobus et al., 2017; Tars and Fishel, 2018), sampling of training data (Pham et al., 2022), and developing multi-head attention networks for each domain (Jiang et al., 2020). These approaches aim to enable a single model to handle various domains.

Instead of a multi-domain model, an expert model for specific domains has also been explored (Neves et al., 2022) to enhance in-domain performance. To address multiple domains, collaborating multiple expert models is a feasible strategy. However, previous studies have yet to examine the

effectiveness of this strategy due to the cost of running multiple models. In light of the growing trend toward larger models, it is valuable to explore the performance potential of collaborative (smaller) models.

**Collaboration of multiple models.** Collaboration methods for multiple models (ensemble) have been explored. The typical method entails averaging the probabilities of each output token across multiple models. Kobayashi (2018) proposed the post-ensemble, which compares the outputs from multiple models and selects the one closest to the majority among the outputs. Ensemble methods are often applied to multiple models trained on the same data but with different random seeds, whereas we perform ensembles on models trained on different domains. Furthermore, output selection (or re-ranking), which selects the best one from outputs, can benefit DEMs. Fernandes et al. (2022) investigated the effects of output selection using minimum bayes risk decoding (Kumar and Byrne, 2004) and quality estimation methods (Kepler et al., 2019; Ranasinghe et al., 2020) in MT task. However, their experiment only examines the effect of output selection when a single model has multiple outputs. We investigate the gain when these ensemble and output selection methods are applied to multiple expert models.

## 3 Investigation Focus

The multi-domain translation task assumes $n$ domains, with data $D_i (i \in \{1, 2, ..., n\})$ corresponding to each domain. Besides the domain data, general domain data $D_{general}$ is assumed to be available. Based on the existing domain adaptation approach (Luong and Manning, 2015; Sennrich et al., 2016), the model is pre-trained in $D_{general}$, then fine-tuned in the domain data. At test time, no domain information is provided with the source, although it is an instance of one of the $n$-domains. Following is a description of MDMs and DEMs.

**Multi-Domain Model (MDM).** In MDM, a single translation model is pre-trained on $D_{general}$, followed by fine-tuning with domain-mixed data $D_{all} = D_1 + ... + D_n$.

**Domain Expert Models (DEMs).** In DEMs, a single model is pre-trained on $D_{general}$, as in MDM setting. Then, the pre-trained model is fine-tuned as an expert model on each domain data $D_i$.

| Dataset | #Sent Pairs |
|---|---|
| JParaCrawl v3.0 | 25.7M |
| The Kyoto Free Translation Task (KFTT) | 440k |
| Japanese-English Legal Parallel Corpus (LAW) | 260k |
| TED talks (TED) | 225k |
| Asian Scientific Paper Excerpt Corpus (ASPEC) | 200k |
| The Business Scene Dialogue corpus (BSD) | 20k |

Table 1: Number of parallel sentences in training data.

Thus, $n$ expert models are developed. At inference time, either an ensemble is applied, or all expert models generate each output, and then the output selection algorithm determines the final output. Section 4.2 describes the output selection and ensemble methods used in our experiments.

## 4 Experiments

We performed experiments on English-to-Japanese (En-Ja) and Japanese-to-English (Ja-En) translations with five domains.

### 4.1 Datasets, models, and evaluation

As shown in Table 1, we used JParaCrawl v3.0 (Morishita et al., 2022) for pre-training and the following five datasets for fine-tuning: The Kyoto Free Translation Task (KFTT) (Neubig, 2011), Japanese-English Legal Parallel Corpus (LAW)[2], TED talks (TED) (Cettolo et al., 2012), Asian Scientific Paper Excerpt Corpus (ASPEC) (Nakazawa et al., 2016), and The Business Scene Dialogue corpus (BSD) (Rikters et al., 2019)[3]. The concept of 'domain' can be defined from several perspectives (Saunders, 2022). In this study, we equated differences in datasets with variations in domains. Thus, in the DEMs setting, we fine-tuned five models as the domain experts from the identical pre-trained model.

We built transformer-based MT models (Vaswani et al., 2017) of three different sizes: SMALL (90M), BASE (290M), and LARGE (1B). The differences in model configurations for each size are shown in Table 2. To make a fair comparison in both MDM and DEMs settings, (i) for pre-training, the same number of epochs and batch sizes were used for both settings, and (ii) for fine-tuning, MDM was updated 10k times, and each expert model in DEMs was updated 2k times, with the same batch size.[4]

---

[2]http://www.phontron.com/jaen-law/
[3]Appendix A shows the details of the datasets.
[4]Appendix B shows the details of the models.

| Name | Params | Encoder & Decoder | | | |
|---|---|---|---|---|---|
| | | layers | $d_{model}$ | $d_{ffn}$ | heads |
| SMALL | 90M | 6 | 512 | 2048 | 8 |
| BASE | 290M | 6 | 1024 | 4096 | 16 |
| LARGE | 1B | 6 | 2048 | 8192 | 32 |

Table 2: Model Configurations.

We chose MS-COMET-22 (Kocmi et al., 2022) as our primary evaluation metric. Since MS-COMET-22 is an evaluation model developed by additionally training COMET (Rei et al., 2020) with a variety of domain data, it is considered suitable for a multi-domain setting.

## 4.2 Output selection and Ensemble for DEMs

Let $y^i$ represent the output of the $i$-th expert model, and let $\mathbb{Y}$ be the set of $y^i$ for all $i$. The following explains the selection method for the final outputs in the DEMs setting;

**Quality Estimation (QE):** Select one from $\mathbb{Y}$ with the highest score of the quality estimation metric, MS-COMET-QE-22 (Kocmi et al., 2022).

**Minimum Bayes Risk (MBR):** Select one as follows, $y_{\text{MBR}} = \text{argmax}_{y^i \in \mathbb{Y}} \frac{1}{|\mathbb{Y}|} \sum_{j=1}^{|\mathbb{Y}|} u(x, y^j, y^i)$, where $u$ is MS-COMET-22, a reference-based metric measuring the similarity between a hypothesis $y \in \mathbb{Y}$ and a pseudo reference $y^* \in \mathbb{Y}$, using also information of source $x$.

**Ensemble (ENS):** The token probability is calculated as follows, $p(y_{\leq t}|x) = \frac{1}{|\mathbb{Y}|} \sum_{j=1}^{|\mathbb{Y}|} p_j(y_{\leq t}|x)$, where $p_j$ denotes the token probability of the expert model, and $x$ represents the input. Subsequently, beam search is employed to generate the output.

**Domain Match (DM):** Manually select one according to the domain of the input. This is a cheat method because the domain information is generally unknown, as Section 3 described.

**Oracle (Oracle):** Select one with the highest score according to the MS-COMET-22, taking the reference into account as well. This method yields an upper bound in the DEMs setting.

QE, MBR, DM, and Oracle are the output selection methods. Given that QE and MBR require only the output of each model, and ENS can be performed if the weights of each model are accessible, QE, MBR, and ENS are practical settings for real-world applications. In contrast, DM needs to obtain domain information for input data, and Oracle needs to access references, so DM and Oracle are experimental settings.

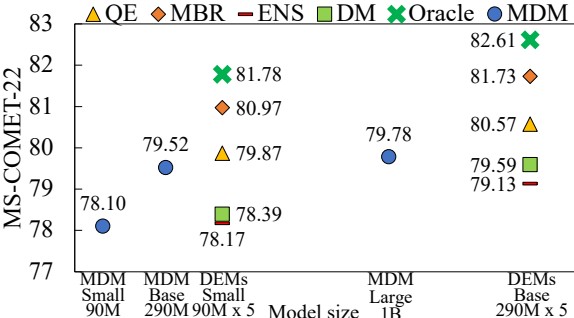

Figure 2: Evaluation results for En-Ja test data.

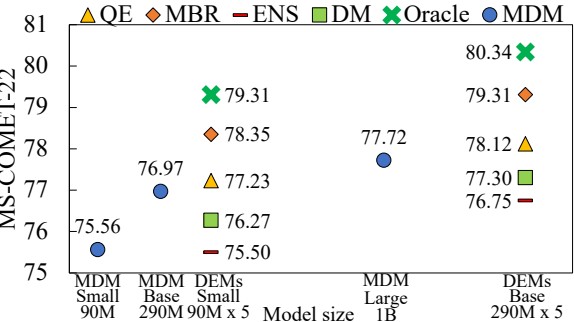

Figure 3: Evaluation results for Ja-En test data.

## 5 Results, Analyses and Discussions

Figures 2 and 3 show the results for the En-Ja and Ja-En test data, respectively.[5] Comparing MDM and DEMs for the same model size, DEMs scored higher for almost all methods. In addition, DEMs-SMALL (90M × 5) achieved competitive performance to MDM-LARGE (1B) by Oracle and MBR. We assessed statistical significance through the paired t-test and bootstrap resampling (Koehn, 2004). The results indicated that Oracle and MBR outperformed compared to MDM settings. These findings suggest that effective combinations of small expert MT models can surpass the performance of a larger MT model.

**Comparison of output selection methods and Ensemble for DEMs:** Across all model sizes and language directions, the five methods ranked Oracle > MBR > QE > DM > ENS. The difference between QE and Oracle can be ascribed to the performance gap between MS-COMET-QE-22 and MS-COMET-22. It seems counter-intuitive that MBR scored higher than QE. Considering the superior translation performance of the expert model in the target domain, it is reasonable to expect that QE,

---

[5]Appendix C shows the results for metrics other than MS-COMET-22. Appendix D shows the results across all model and domain combinations. Appendix E shows results on development data. Appendix F shows examples of DEMs output.

| Data\Model | KFTT | LAW | TED | ASPEC | BSD |
|---|---|---|---|---|---|
| KFTT | **155** | 92 | 102 | 118 | 128 |
| LAW | 97 | **260** | 97 | 106 | 101 |
| TED | 105 | 104 | 112 | 66 | **132** |
| ASPEC | 99 | 100 | 79 | **134** | 101 |
| BSD | 98 | 116 | 105 | 100 | **179** |

Table 3: Confusion matrix of domains selected by DEMs (`Oracle`) in the En-Ja test data. Each number is a count. If multiple models get the same best score, they all get a count.

| Data\Model | KFTT | LAW | TED | ASPEC | BSD |
|---|---|---|---|---|---|
| KFTT | **127** | 120 | 117 | 105 | 123 |
| LAW | 119 | **211** | 133 | 96 | 117 |
| TED | 95 | 141 | 76 | 61 | **151** |
| ASPEC | 127 | 115 | 87 | 33 | **149** |
| BSD | 71 | 134 | 104 | 80 | **195** |

Table 4: Confusion matrix of domains selected by DEMs (`QE`) in the En-Ja test data. Each number is a count. If multiple models get the same best score, they all get a count.

| Data\Model | KFTT | LAW | TED | ASPEC | BSD |
|---|---|---|---|---|---|
| KFTT | 128 | 86 | 111 | 136 | **160** |
| LAW | 148 | 133 | 151 | **179** | 141 |
| TED | **152** | 130 | 70 | 84 | 105 |
| ASPEC | **135** | 112 | 77 | 73 | 127 |
| BSD | 153 | 155 | 95 | **158** | 99 |

Table 5: Confusion matrix of domains selected by DEMs (`MBR`) in the En-Ja test data. Each number is a count. If multiple models get the same best score, they all get a count.

which selects a single high-quality output, would surpass MBR, which chooses the sentence most semantically similar to others in $\mathbb{Y}$. Therefore, improving QE methods is crucial for the success of DEMs setting. Even the expert model that matches the input domain can generate low-quality output. In such cases, MBR and QE can select better output from other models. This may account for the DM results being lower than the MBR and QE.

ENS scored lower than QE and MBR. This result suggests that in the DEMs setting, ENS is negatively impacted by models with poor performance. Hence, weighted ENS could improve the performance (Durrani et al., 2016; Sajjad et al., 2017). However, determining the appropriate weights typically necessitates training or tuning, and any alteration in the number of domains requires a reconfiguration of these weights. QE and MBR would be more flexible than the ENS in multi-domain MT tasks.

**Selected outputs in `Oracle`, `QE`, and `MBR`:** Tables 3, 4, and 5 show which expert model's outputs were selected by `Oracle`, `QE`, and `MBR` for each domain of input in En-Ja, respectively. For `Oracle`, the corresponding expert model's output for the input domain was consistently the most selected, but this was not the case for MBR and QE. Even in `Oracle`, outputs from models other than the expert one corresponding to the input were often selected. This suggests that the DM did not lead to high performance. We assume that this is an effect of including instances whose domain is not far distant from the different domain data. Since `Oracle` simply selects good-quality sentences without being aware of the domain, it can select the appropriate model's output flexibly. This appears to explain the significant difference in scores between `Oracle` and DM.

**Effects of multiple models and data separating:** As discussed in the previous paragraph, we can assume that there are two contributing factors to the

performance superiority of DEMs: 1) the use of multiple models, and 2) domain-based data separation. To verify these two factors separately, we prepared five MDMs trained with various random seeds. We then evaluated the performance of QE, MBR, ENS, and `Oracle`, which are the identical methods used for DEMs. Note that the initial model for fine-tuning is identical across all models in both DEMs and MDMs, with differences emerging solely in the fine-tuning data. Table 6 shows the performance of DEMs and MDMs on En-Ja test data.[6]

In MDMs, the average score (AVG) of the five models was lower than all other methods (QE, MBR, ENS, `Oracle`). This indicates that the use of multiple models is effective. Also, the output selection methods (QE and MBR) scores were higher than ENS. This result suggests that the output selection approach is more effective than the typical ensemble approach, even when each model is similar (since the MDMs have the same training data and differ only in seed).

Comparing DEMs and MDMs, it is intriguing to note that DEMs outperformed MDMs in all methods except AVG, even though DEMs uses less data for fine-tuning than MDMs. In addition, none

---

[6]Each model in MDMs was fine-tuned one epoch for each model (about 1.5 times the number of updates for each model in DEMs). Appendix G shows the results when training a model in MDMs setting with different epochs.

|        | AVG   | QE    | MBR   | ENS   | Oracle |
|--------|-------|-------|-------|-------|--------|
| DEMs   | 77.20 | **79.87** | **80.97** | **78.17** | **81.78** |
| MDMs   | **78.15** | 78.61 | 78.63 | 78.15 | 78.97 |
| diff   | -0.95 | +1.26 | +2.34 | +0.02 | +2.81 |

Table 6: Performance comparison of DEMs and MDMs with SMALL on En-Ja test data. The 'diff' row is the DEMs score minus the MDMs score. The scores of DEMs are the same as the DEMs-SMALL 90M x 5 in Figure 2.

of the methods for MDMs exceeded the MDM-LARGE score (79.78) in Figure 2. These results suggest that domain-based data separation is effective. The difference between DEMs and MDMs scores for the output selection methods (QE and MBR) stems from a heightened probability of including good outputs in the candidate set. This is because DEMs can more dependably acquire domain-specific knowledge through domain-based data separation.

**Effects of selection model performance on DEMs:** We conducted experiments using wmt22-cometkiwi-da (Rei et al., 2022b) and wmt22-comet-da (Rei et al., 2022a) in place of MS-COMET-QE-22 and MS-COMET-22, respectively. We also used wmt22-comet-da as an evaluation metric. Figures 4 and 5 show the results for the En-Ja and Ja-En test data, respectively.

Compared to the MS-COMET-QE-22 and MS-COMET-22 experiments shown in Figures 2 and 3, QE and MBR showed a smaller increase in scores. This result can be attributed to the fact that MS-COMET-QE-22 and MS-COMET-22 have been trained on multi-domain data, as discussed in Section 4.1, whereas wmt22-cometkiwi-da and wmt22-comet-da are not. This indicates that the metrics used in a multi-domain setting must be multi-domain capable.

**Potential of DEMs in practice:** We believe that DEMs approach offers several advantages over MDM, especially in practical scenarios. For example, DEMs approach provides benefits for multi-organizational collaboration in cases where privacy issues preclude sharing training data. Since when QE and MBR are applied in DEMs, it is enough to share only the model outputs. Although not explored in our experiment, because DEMs would work even with different model architectures, it may facilitate collaboration among multiple organizations compared to Federated Learning (McMa-

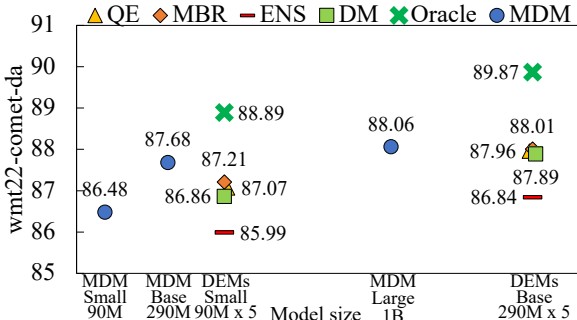

Figure 4: Evaluation results for En-Ja test data. QE was conducted with wmt22-cometkiwi-da, and MBR, Oracle, and evaluation was conducted with wmt22-comet-da.

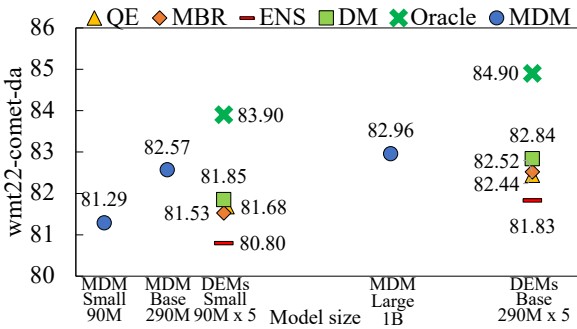

Figure 5: Evaluation results for Ja-En test data. QE was conducted with wmt22-cometkiwi-da, and MBR, Oracle, and evaluation was conducted with wmt22-comet-da.

han et al., 2016; Passban et al., 2022), for example. A more comprehensive study will be conducted in the future.

## 6 Conclusion

We investigated the effectiveness of MT models in a multi-domain setting in fair conditions and explored a direction for tackling multi-domain MT. We mainly compared the performance of the two main approaches, MDM and DEMs, on English to/from Japanese tasks with five domains. Our results revealed that the domain-based data separation strategy is effective in a multi-domain setting. Additionally, we found that when we want to use metrics for selection in a multi-domain setting, the metrics must be multi-domain capable. Furthermore, combining five 90M DEMs based on an output selection approach has the potential to perform better than a single 1B MDM, which may suggest a new direction for developing MT models in the multi-domain setting.

## Limitations

Our experiments were conducted only on the English to/from Japanese translation task. Also, we only experimented with five domains. The number of domains increases, as does the number of expert models, and the running cost of the DEMs increases as well. In addition, both MDM and DEMs approaches can be applied to tasks other than translation. However, in our experiments, DEMs depends on the performance of MS-COMET-22 and MS-COMET-QE-22, and in tasks where such tools are not available, our scenario used may not be applicable. Furthermore, we did not include in our comparisons any studies such as domain tagging, training data sampling, or model architecture that would improve the performance of MDM, as described in Section 2. Experimentation in a more comprehensive setting is future work.

We discussed the trade-off between the performance and size of the models in MDM and DEMs. Model compression methods such as quantization and knowledge distillation could impact the trade-off. However, such compression techniques are outside our focus.

To achieve good results with DEMs settings, there needs to be at least one high-quality candidate among the expert model's outputs. If the performance of expert models is inadequate, the output selection methods discussed might not be as effective.

In our experiments, models were evaluated using only automatic evaluation metrics. While we recognize that human evaluation is necessary to perform a precise evaluation in a translation task, we only performed automatic evaluation due to the large number of models to compare.

## Ethics Statement

Our experiments were conducted only on the datasets already published to the community and well-studied in many previous studies. Moreover, even though our target task is machine translation, which involves text generation, we cannot believe the methods presented here further amplify biases for any other ethical concerns that implicitly exist in the datasets and tasks. Therefore, we foresee no ethical concerns in this work.

## Acknowledgements

We would like to thank the members of the Tohoku NLP Group for their insightful comments. This work was supported by JST Moonshot R&D Grant Number JPMJMS2011 (fundamental research) and JST CREST Grant Number JPMJCR20D2.

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

## A  Details of dataset construction

For fine-tuning on KFTT, TED, and BSD, we used the original training data. The original LAW contains 262,449 instances without distinction between train, development, and test, so we used all but the first 2,000 instances for fine-tuning. The original ASPEC contains 3,008,500 instances in the training set, but we used only the first 200,000 instances for fine-tuning.

We created the development and test data by concatenating the five datasets used for fine-tuning: KFTT, LAW, TED, ASPEC, and BSD. They were each extracted from the top 500 instances after duplicates were removed from each raw data. Consequently, there were 2,500 unduplicated instances each for development and test data. For the raw data on KFTT, ASPEC, and BSD, we used the original development and test data. In the case of the raw data on TED, we used dev2010 for development data and tst2015 for test data. For the raw data on LAW, we used the first 1~1000 instances of the original data for development data and the next 1001~2000 instances for test data.

We tokenized all data into subwords as a pre-processing step. We trained subword segmentation models using byte pair encoding on the JParaCrawl v3.0 as the training dataset for English and Japanese, respectively. In training, we applied byte fallback, ensuring essentially the absence of unknown words in the language used for any data, including fine-tuning data. We chose a vocabulary size of 32k for each language. We used `Sentencepiece` tool (Kudo and Richardson, 2018) for these implementations.

## B  Details of model configurations

The basic model configuration was the *transformer* defined in `fairseq` toolkit (Ott et al., 2019). From the default settings, we moved the layer normalization to the beginning of each transformer block, made sharing the input and output embedding of the decoder, and implemented the cross+self-attention mechanism (Peitz et al., 2019).

We trained neural MT models with the `fairseq` tool in version 0.12.2. In pre-training, we used the same configuration for SMALL, BASE, and LARGE except for the learning rate (See Table 7). In fine-tuning, we used the same configuration for SMALL, BASE, and LARGE (See Table 8).

We used the same generation configuration for SMALL, BASE, and LARGE, with beam search de-coding, a beam size of 5, and a maximum length of 200.

## C  Results of performance comparison by multiple metrics

Tables 9 and 10 show the results with MS-COMET-22, BLEU (Papineni et al., 2002), chrF (Popović, 2015), and BLEURT (Sellam et al., 2020) for the En-Ja and Ja-En test data. We use sacreBLEU (Post, 2018) for the BLEU and chrF calculations. The sacreBLEU signatures are `nrefs:1|case:mixed|eff:no|tok:13a|smooth :exp|version:2.3.1` and `nrefs:1|case:mixed|eff:yes|nc:6|nw:0|spa ce:no|version:2.3.1` respectively. We used BLEURT-20 for BLEURT.

## D  Results for all combinations of domains and models

Tables 11, 12, 13 and 14 show the results with MS-COMET-22 for all combinations of data and models.

## E  Results of MS-COMET-22 for development data

Figure 6 shows the evaluation results by MS-COMET-22 for the En-Ja development data, and Figure 7 shows the evaluation results by MS-COMET-22 for the Ja-En development data. For both En-Ja and Ja-En, the results for the development data showed a similar trend to the results for the test data.

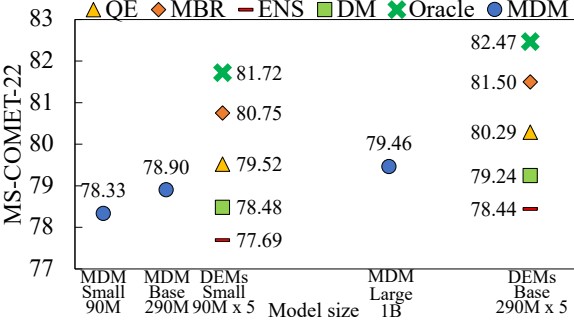

Figure 6: Evaluation results for En-Ja development data.

## F  Examples of outputs for each expert model and output selection

Table 15 shows examples of each expert model's output and the result of output selections for the TED test data in En-Ja.

| Configurations | Selected Value |
|---|---|
| Optimizer | Adam ($\beta_1 = 0.9$, $\beta_2 = 0.98$, $\epsilon = 1 \times 10^{-8}$) |
| Weight Decay | 0 |
| Criterion | label smoothed cross entropy |
| Label Smoothing | 0.1 |
| Learning Rate | SMALL: 0.001, BASE: 0.0005, LARGE: 0.00025 |
| Learning Rate Schedule | inverse square root |
| Warmup Steps | 30000 |
| Gradient Clipping | 1.0 |
| Dropout | 0.1 |
| Mini-batch Size | 512k |
| Number of Epochs | 50 |
| Averaging | Save checkpoint for every 5 epochs and take an average of last 5 checkpoints |

Table 7: List of hyperparameters in pre-training.

| Configurations | Selected Value |
|---|---|
| Optimizer | Adam ($\beta_1 = 0.9$, $\beta_2 = 0.98$, $\epsilon = 1 \times 10^{-8}$) |
| Weight Decay | 0 |
| Criterion | label smoothed cross entropy |
| Label Smoothing | 0.1 |
| Learning Rate | 0.00001 |
| Learning Rate Schedule | constant |
| Gradient Clipping | 1.0 |
| Dropout | 0.2 |
| Mini-batch Size | 16k |
| Number of Updates | MDM: 10000, DEMs: 2000 |
| Averaging | Save checkpoint for every 100 updates and take an average of last 5 checkpoints |

Table 8: List of hyperparameters in fine-tuning.

| | | COMET* | BLEU | chrF | BLEURT |
|---|---|---|---|---|---|
| DEMs-SMALL | QE | 79.87 | 24.1 | 31.3 | 65.62 |
| | MBR | 80.97 | 22.3 | 30.1 | 65.05 |
| | ENS | 78.17 | 24.6 | 31.6 | 65.55 |
| | DM | 78.39 | 30.6 | 36.3 | 67.39 |
| | Oracle | 81.78 | 25.8 | 32.7 | 66.94 |
| DEMs-BASE | QE | 80.57 | 25.7 | 32.9 | 66.98 |
| | MBR | 81.73 | 24.4 | 32.0 | 66.42 |
| | ENS | 79.13 | 26.9 | 33.8 | 67.24 |
| | DM | 79.59 | 33.0 | 38.5 | 69.28 |
| | Oracle | 82.61 | 28.2 | 35.0 | 68.94 |
| MDM-SMALL | | 78.10 | 29.8 | 35.6 | 66.18 |
| MDM-BASE | | 79.52 | 32.5 | 38.0 | 68.41 |
| MDM-Large | | 79.78 | 34.1 | 39.4 | 69.30 |

Table 9: Performance comparison of all settings on En-Ja test data using metrics COMET* (MS-COMET-22), BLEU, chrF, and BLEURT.

| | | COMET* | BLEU | chrF | BLEURT |
|---|---|---|---|---|---|
| DEMs-SMALL | QE | 77.23 | 22.4 | 52.2 | 66.02 |
| | MBR | 78.35 | 22.5 | 52.0 | 65.98 |
| | ENS | 75.50 | 23.6 | 52.5 | 66.11 |
| | DM | 76.27 | 28.1 | 55.8 | 67.76 |
| | Oracle | 79.31 | 25.0 | 53.9 | 67.88 |
| DEMs-BASE | QE | 78.12 | 23.5 | 53.6 | 67.25 |
| | MBR | 79.31 | 24.0 | 53.6 | 67.57 |
| | ENS | 76.75 | 25.3 | 54.2 | 67.57 |
| | DM | 77.30 | 30.3 | 58.0 | 69.52 |
| | Oracle | 80.34 | 27.0 | 55.8 | 69.82 |
| MDM-SMALL | | 75.56 | 27.8 | 55.8 | 67.27 |
| MDM-BASE | | 76.97 | 30.0 | 57.7 | 69.25 |
| MDM-Large | | 77.72 | 30.6 | 58.5 | 69.92 |

Table 10: Performance comparison of all settings on Ja-En test data using metrics COMET* (MS-COMET-22), BLEU, chrF, and BLEURT.

# G  Whether the MDMs models are sufficiently trained

We investigate the impact of the number of training epochs on the performance of the MDMs in the section 5 experimental setting. Table 16 shows the translation performance with one epoch, two epoch, and three epoch fine-tuning for the concatenated data of KFTT, LAW, TED, ASPEC and BSD. The epoch-1 demonstrated the highest performance, and additional increments in the number of epochs did not enhance the translation performance. Therefore, the MDMs model did not appear to be undertrained, and it is more likely that domain-based separated data is more effective than multi-domain data for training.

| Data\Model | MDM | KFTT | LAW | TED | ASPEC | BSD |
|---|---|---|---|---|---|---|
| MIX | **78.10** | -1.08 | -0.70 | -1.62 | -1.11 | -0.02 |
| KFTT | **77.06** | -0.89 | -2.37 | -2.58 | -1.37 | -0.30 |
| LAW | 81.51 | -2.54 | **+0.86** | -4.51 | -2.98 | -1.95 |
| TED | 73.31 | -0.80 | -0.35 | +0.13 | -0.57 | **+0.86** |
| ASPEC | 80.17 | -0.16 | -1.01 | -1.46 | **+0.27** | +0.21 |
| BSD | 78.47 | -1.01 | -0.60 | +0.34 | -0.86 | **+1.08** |

Table 11: Results for all combinations of data and models with SMALL on En-Ja test data. In each row, the score of each expert model is listed based on the MDM score.

| Data\Model | MDM | KFTT | LAW | TED | ASPEC | BSD |
|---|---|---|---|---|---|---|
| MIX | **79.52** | -1.39 | -1.53 | -1.86 | -1.49 | -0.77 |
| KFTT | 78.14 | **+0.09** | -1.93 | -1.92 | -0.28 | -1.16 |
| LAW | **83.01** | -2.00 | -0.35 | -5.83 | -2.66 | -2.40 |
| TED | 75.70 | -1.86 | -2.04 | **+0.10** | -1.75 | -0.50 |
| ASPEC | 80.98 | -0.91 | -1.43 | -1.54 | **+0.02** | -0.30 |
| BSD | 79.76 | -2.26 | -1.89 | -0.13 | -2.77 | **+0.51** |

Table 12: Results for all combinations of data and models with BASE on En-Ja test data. In each row, the score of each expert model is listed based on the MDM score.

| Data\Model | MDM | KFTT | LAW | TED | ASPEC | BSD |
|---|---|---|---|---|---|---|
| MIX | **75.56** | -0.24 | -0.18 | -2.13 | -1.65 | -0.21 |
| KFTT | **71.03** | -0.02 | -2.04 | -5.96 | -2.45 | -2.33 |
| LAW | 78.73 | -1.41 | **+0.45** | -2.39 | -2.71 | -1.33 |
| TED | 73.67 | -0.39 | -0.15 | **+0.22** | -1.41 | +0.07 |
| ASPEC | 79.05 | -0.03 | **+0.07** | -1.10 | -0.09 | -0.43 |
| BSD | 75.33 | +0.66 | +0.77 | -1.44 | -1.58 | **+2.96** |

Table 13: Results for all combinations of data and models with SMALL on Ja-En test data. In each row, the score of each expert model is listed based on the MDM score.

| Data\Model | MDM | KFTT | LAW | TED | ASPEC | BSD |
|---|---|---|---|---|---|---|
| MIX | **76.97** | -0.46 | -0.66 | -2.56 | -2.11 | -0.30 |
| KFTT | **73.27** | -0.35 | -2.09 | -6.06 | -2.48 | -2.38 |
| LAW | 79.57 | -1.23 | **+0.29** | -3.13 | -3.06 | -1.06 |
| TED | 75.03 | -0.24 | -0.80 | +0.21 | -1.98 | **+0.55** |
| ASPEC | 79.59 | 0.00 | **+0.10** | -1.33 | -0.26 | -0.38 |
| BSD | 77.38 | -0.46 | -0.83 | -2.49 | -2.80 | **+1.78** |

Table 14: Results for all combinations of data and models with BASE on Ja-En test data. In each row, the score of each expert model is listed based on the MDM score.

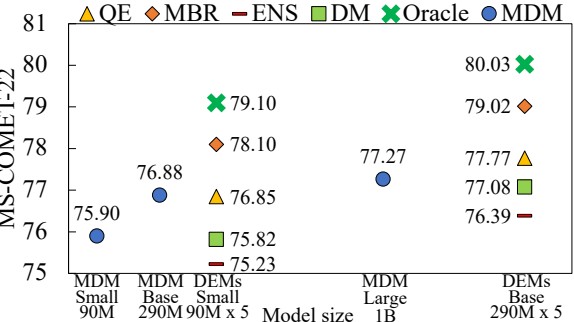

Figure 7: Evaluation results for Ja-En development data.

| | | |
|---|---|---|
| source | リビングストン公立図書館は私の故郷で2004年に完成しました
ドームをたずさえ丸い装飾と円柱に赤レンガときました
リビングストンがこの建物で何を伝えたいのか検討がつくでしょう
子供たち、資産価値、歴史です | |
| reference | This is the Livingston Public Library that was completed in 2004 in my hometown, and, you know, it's got a dome and it's got this round thing and columns, red brick, and you can kind of guess what Livingston is trying to say with this building: children, property values and history. | |
| KFTT model's output | The Livingstone Public Library was completed in 2004 in my hometown. | |
| LAW model's output | The Livingstone Public Library was completed in 2004 in my hometown,
and the Livingstone Public Library was completed in 2004.
The Livingstone Public Library was completed in 2004.
The Livingstone Public Library was completed in 2004. | |
| TED model's output
(selected by MBR, DM, Oracle) | The Livingstone Public Library was completed in 2004 in my hometown,
with a dome, a round decoration, a column, red brick,
and you're going to see what Livingstone wants to tell you about in this building:
children, asset value, history. | |
| ASPEC model's output | Livingstone Public Library was completed in my hometown in 2004. | |
| BSD model's output
(selected by QE) | The Livingstone Public Library was completed in 2004 in my hometown,
and it's a dome with round decorations and columns with red bricks
Livingstone will be able to consider what you want to tell this building
Children, asset value, history | |

Table 15: Examples of DEMs-SMALL's output. The source and reference are sampled from the TED dataset. The MBR, MD, and Oracle methods selected the expert model fine-tuned on TED. The QE method selected the output of the expert model fine-tuned on BSD.

| | AVG | QE | MBR | ENS | Oracle |
|---|---|---|---|---|---|
| epoch-1 | **78.15** | **78.61** | **78.63** | **78.15** | 78.97 |
| epoch-2 | -0.23 | -0.23 | -0.14 | -0.14 | -0.13 |
| epoch-3 | -0.12 | -0.08 | -0.02 | -0.09 | **+0.04** |

Table 16: Performance comparison of SMALL with varying training epochs in MDMs on En-Ja test data.