# OpenReview forum: "Investigating the Effectiveness of Multiple Expert Models Collaboration"
_EMNLP/2023/Conference — EMNLP 2023 Findings_

### Official Review · Reviewer_Va6z · 2023-08-03

**Soundness:** 3

**Excitement:**

3: Ambivalent: It has merits (e.g., it reports state-of-the-art results, the idea is nice), but there are key weaknesses (e.g., it describes incremental work), and it can significantly benefit from another round of revision. However, I won't object to accepting it if my co-reviewers champion it.

**Paper Topic And Main Contributions:**

The focus of this paper is to compare the approach of training a single machine translation (MT) model that covers all domains and the approach that aggregates models of multiple domains under a fair condition. The authors conduct experiments on multiple domain datasets and demonstrate that a combination of smaller Domain Expert Models (DEMs) can outperform a single larger Multi-Domain Model(MDM).

**Questions For The Authors:**

If we selected different outputs of the MDM model using the MBR method,  can the MDM method achieve higher performance than DEMs?
This paper only compares two multiple-domain settings under English-Japanese translation tasks, which are dissimilar between the two languages. I wonder if we conduct experiments on similar languages,  can we achieve the same conclusion as the En-Ja translation task?

**Reasons To Accept:**

This paper sheds light on the underexplored domain expert models (DEMs) and verifies the hypothesis that DEMs have the potential to achieve superior performance if each expert model is well-trained for the corresponding domain and properly combined. This paper conducts experiments on different output selection approaches and shows under what method DEMs outperform MDMs.

**Reasons To Reject:**

The authors only conduct experiments on En-Ja translation tasks, which is not sufficient to support the conclusion in this paper.
In the DEMs method, the output of different models should be selected or ensembled in the postprocess period, which is not suitable in the actual NMT system.
To acquire results of QE and Oracle methods, the external evaluation model or references would be relied on. I do not think it is a fair comparison.

**Reproducibility:**

5: Could easily reproduce the results.

**Reviewer Confidence:**

5: Positive that my evaluation is correct. I read the paper very carefully and I am very familiar with related work.

---

> ### Author Rebuttal · Authors · 2023-08-29
>
> Thank you for the thoughtful review.
> ## About the practicality of DEMs settings
> We believe that the DEMs (Domain Expert Models) approach is suitable for the real MT systems.
>
> (i) As Reviewer #Va6z mentioned, QE (Quality Estimation) and MBR (Minimum Mayes Risk) require external evaluation models. However, evaluation metrics such as COMET are available as open source, and access to them is easy. Also, for example, the use of external evaluation models is allowed in the WMT competition (https://www2.statmt.org/wmt23/translation-task.html).
>
> (ii) During inference, in a DEMs setting, each output of the expert model can be generated independently and can easily be processed in parallel. So, for example, DEMs with each model being small would be able to be inferred on multiple small-memory GPU machines. Although the appropriate approach will vary depending on the situation of computational resources and other factors, DEMs would be useful in a real-world setting.
>
> (iii) The DEMs setting is one of the most common approaches in shared tasks such as WMT translation (Kiyono et al., WMT 2020).
>
> (iv) We believe DEMs would be beneficial in a multi-organizational collaboration scenario, as discussed in lines:260-271.
>
> We add these discussions in the camera-ready version.
>
> - [Tohoku-AIP-NTT at WMT 2020 News Translation Task](https://aclanthology.org/2020.wmt-1.12) (Kiyono et al., WMT 2020)
> ## About the appropriateness of using an external evaluation model in a fair comparison of MDM and DEMs
> Oracle is a setting to know the upper bound in the DEMs approach, not a baseline for fair comparisons. Although QE and MBR require an external evaluation model, our research question is to investigate the performance improvement of those evaluation models in the DEMs setting. We believe that our experiment setting is a fair comparison for our research questions.
>
> ## Applying MBR to MDM
> It is expected that applying MBR to MDM (a Multi-Domain Model) will also improve performance (Fernandes et al., NAACL 2022). Also for DEMs, increasing the number of hypotheses generated by each model may improve performance. Note that our experiment was conducted in a setting where each model has a single hypothesis for both MDM and DEMs.
>
> - [Quality-Aware Decoding for Neural Machine Translation](https://aclanthology.org/2022.naacl-main.100) (Fernandes et al., NAACL 2022)
> ## About experiments with more language pairs
> As discussed in the limitation, we also recognize that the fact that the language pairs are only English from/to Japanese is a limitation of our paper. There is no doubt that our paper would be better if we experimented with more language pairs.
> However, we believe that our experimental setting is well designed for the research question and that our paper provides a useful case study because English from/to Japanese is one of the major language pairs in the translation task, which is included in the WMT General (News) translation task. We also believe that we have provided sufficient evidence to support our claims in light of the review policy for a short paper (https://aclrollingreview.org/reviewertutorial#4-how-to-read-for-review).

---

### Official Review · Reviewer_6uVB · 2023-08-03

**Soundness:** 3

**Excitement:**

2: Mediocre: This paper makes marginal contributions (vs non-contemporaneous work), so I would rather not see it in the conference.

**Missing References:**

Very complete overview of recent related work by Saunders 2022 http://arxiv.org/abs/2104.06951
Pham et al, 2022 https://aclanthology.org/2022.eamt-1.4.pdf
Pham et al 2021 https://direct.mit.edu/tacl/article/doi/10.1162/tacl_a_00351/97775/Revisiting-Multi-Domain-Machine-Translation
Gu, Feng 2020 https://arxiv.org/pdf/2011.00678.pdf

**Paper Topic And Main Contributions:**

This paper addresses the problem of domain-specific Machine Translation, and is specifically interested in the scenario when we have access to different domains data during training, but during inference the input domain is unknown. It compares  (1) default multi-domain model setting, where single model is trained on the concatenation of all the datasets coming from differeint domains and (2) different ensembling methods combining several (smaller) domain-specific models. Domain specific models are trained in 2 steps: first general model is trained on concatenation of all the data, then general model is finetuned on the domain of interest. The ensembling methods this work is comparing are
1) Oracle: simply selecting the translation with highes MS-COMET-22 score
2) QE: selecting the translation with highes MS-COMET-QE (referenceless) score
3) MBR: minimum bayes risk optimisation relying on MS-COME-22 estimator
4) ENS: each token probablilty is an avg probablity across all domain-specific models
5) DM : provide model with golden domain and select appropriate domain during inference

Authors report results in MS-COMET-22 scores for en-ja and ja-en datasets containing 5 different domains and demonstrate that Oracle > MBR > QE > DM > ENS

To summarize, looks like that methods that have direct access to the final evaluation metric (Oracle, MBR) outperform the  methods that aim to optimize different metric (which is not that surprising in itself). The face that DM is lower than QE probably indicates that the notion of the domain is not very well defined ( as highligheted by Saunders 2022), and it is not enough to provide model with golden domain label to obtain optimal translation.

**Questions For The Authors:**

line 189 : you seem to be suprised by MBR outperforming QE, but according to line 156 y_MBR does optimise MS-COMET-22, which is your evalaution metric ( not sure how fair this comparison to other methods, which may not have access to the reference)

**Reasons To Accept:**

- Paper is clearly written, experimental choices are well documented
- Authors compare different ensembling strategies which could be of interest to some researchers
 - The finding that QE could be an interesting strategy for ensembling several models seems interesting (if the significance of these results is confirmed)

**Reasons To Reject:**

IMO several important baselines are missing :
  - model with adapter layers seems  (and AdapterFusion ) seem to be a natural baseline in these settings
  - we know that naive concatenation of different domains is subpoptimal baseline, several works have tried different sampling techniques for different domains (Eg. Pham et al. EAMT 2022) , it would have been useful at least to mention such works, and even better to use them as stronger baselines
 - one very simple baseline would be to train a classifier that would be able to predict the domain for input sentence, and report it as a more realistic alternative to DM version

Significance testing would be useful to asses the gains. Moreover, reporting the results in terms of other metrics (Eg. BLEU, chrf, BLEURT?) would have been helpful to asses the validity of the findings

**Reproducibility:**

3: Could reproduce the results with some difficulty. The settings of parameters are underspecified or subjectively determined; the training/evaluation data are not widely available.

**Reviewer Confidence:**

4: Quite sure. I tried to check the important points carefully. It's unlikely, though conceivable, that I missed something that should affect my ratings.

---

> ### Author Rebuttal · Authors · 2023-08-29
>
> We appreciate your constructive comments.
>
> ## About the Minimum Mayes Risk (MBR) algorithm
> We are afraid to say Reviewer #6uVB may have misunderstood that the computation of MBR needs the references. __Minimum Mayes Risk (MBR) uses reference-based metrics such as COMET, but does not require access to reference text, as described in Section 4.2 (line 156). Therefore, MBR is also a "realistic" approach.__
> Furthermore, as shown in Figure 2-5, the MBR achieves 90M x 5 DEMs (Domain Expert Models) outperforming 1B MDM (a Multi-Domain Model).
>
> In the camera-ready version, we will improve the presentation to specify that MBR is the method that does not use reference texts. Moreover, we will clearly state that the practical methods are MBR, QE (Quality Estimation), and ENS (Ensemble), and that the virtual settings are DM (Domain Match) and Oracle.
>
> We would appreciate it if you could reconsider the scores If misunderstanding about MBR primarily affected the decision of the scores.
>
> ## About domain settings
> As reviewer #6uVB mentioned, the domain is not very well defined in our experiment, and we agree that this is one of the reasons why DM is lower than QE and MBR. Domain is an arbitrary delimitation, and it is not easy to separate it exactly, and based on practical situations. Therefore, for convenience, we considered each dataset a different domain, which we believe is a very realistic setting. We will highlight these discussions in the camera-ready version.
>
> ## About baseline settings
> Thank you for the suggestion of many related works. We will add these in the related work section in the camera-ready version. As discussed in the limitation section, research on multi-domains is a very active research area, and we recognize that various approaches can improve MDM performance even more.  There would also be other approaches to DEMs, such as combining classifiers that predict domain, as you suggested. However, these comparisons are outside the focus of this study.
>
> Our research question is to compare output selection and ensemble strategies in a simple setting where the main experimental variables are model size and number of models.
>
> ## Significance test
> We calculated the statistical significance with Paired T-Test and bootstrap resampling (Koehn, EMNLP 2004), using the COMET library implementation.
> The following tables show that MBR is significantly better than MDM.
>
> > If system_x is better than system_y then:
> Null hypothesis rejected according to t-test with p_value=0.05.
> Scores differ significantly across samples.
>
>
> | system_x \ system_y | QE | MBR | ENS | DM | Oracle | MDM-Small (90M) | MDM-Base (290M) | MDM-Large (1B) |
> |----|----|----|----|----|----|----|----|----|
> | MBR (En-Ja) | True | | True | True | False | True | True | True |
> | MBR (Ja-En) | True | | True | True | False | True | True | True |
>
>
> - [Statistical Significance Tests for Machine Translation Evaluation](https://aclanthology.org/W04-3250) (Koehn, EMNLP 2004)
>
> ## Other evaluation metrics
> We do not use BLEU or chrF because they have been reported in recent years to have low evaluation performance (Freitag et al., WMT 2022)  and are likely to mislead to conclusions. We chose MS-COMET as the evaluation metric because MS-COMET has the highest correlation with human judgment (please see the Table 14 and 15 of Freitag et al., WMT 2022). Moreover, MS-COMET is a suitable model for a multi-domain setting, as noted in footnote 5. We believe that MS-COMET is an appropriate metric in our experimental setting, but we will add several additional metrics in the camera-ready version.
>
> - [Results of WMT22 Metrics Shared Task: Stop Using BLEU – Neural Metrics Are Better and More Robust](https://aclanthology.org/2022.wmt-1.2) (Freitag et al., WMT 2022)

---

### Official Review · Reviewer_wUg4 · 2023-08-05

**Soundness:** 3

**Excitement:**

3: Ambivalent: It has merits (e.g., it reports state-of-the-art results, the idea is nice), but there are key weaknesses (e.g., it describes incremental work), and it can significantly benefit from another round of revision. However, I won't object to accepting it if my co-reviewers champion it.

**Paper Topic And Main Contributions:**

The paper compares two model strategies for designing multi-domain systems: multi domain experts and their aggregation vs a single model that is trained on the mixed data. It introduces several ways of aggregating the domain experts. Experiments conducted with 5 domains show that multi experts strategy is a better fit in the scenario concerned in the paper.

Overall, I feel that the paper's result is due to a well-defined scenario: general domain is not a concern in this setup, the 5 domains in experiments seem to be each a focused/concentrated domain. As a result, I wonder if there is a chance that mixing the data in this scenario may not further boot the performance of each -- gradient could even cancel each other if domains are distant. Therefore, I wish that paper provided more discussions and analysis in this regard.

Strengths:
- Paper's writing is clear.
- Thorough experiments to support the final claim.
- The idea proposed in paper will be helpful for practitioners with the same setup.

Weaknesses:
- Wish to see results in more language pairs, and looser (less concentrated domains).
- Examine the failure condition of the winning strategy.


**Reasons To Accept:**

Strengths above: clean, focused examination, results could be practical, interesting (consolidated) ideas for aggregating translations.

**Reasons To Reject:**

Weaknesses above, for example, experiments with more language pairs, with loose domains.

**Reproducibility:**

4: Could mostly reproduce the results, but there may be some variation because of sample variance or minor variations in their interpretation of the protocol or method.

**Reviewer Confidence:**

4: Quite sure. I tried to check the important points carefully. It's unlikely, though conceivable, that I missed something that should affect my ratings.

---

> ### Author Rebuttal · Authors · 2023-08-29
>
> We appreciate your invaluable comments.
>
> ## About experiments with more language pairs and domains
> As discussed in the limitation, we also recognize that the language pairs are only English from/to Japanese and that only five domains are a limitation of our paper.
> However, we believe that our experimental setting is well designed for the research question and that our paper provides a useful case study because English from/to Japanese is one of the major language pairs in the translation task, which is included in the WMT General (News) translation task. We also believe that we have provided sufficient evidence to support our claims in light of the review policy for a short paper (https://aclrollingreview.org/reviewertutorial#4-how-to-read-for-review).
>
>
> ## About examining the failure conditions of the winning strategy
>
> Thank you for your valuable comments.
>
> Due to the nature of the Minimum Mayes Risk (MBR) algorithm, the inclusion of poor quality translations in the candidate set may result in a poor final score.
>
> In the camera-ready version, we will add discussions by showing examples of instances where MBR failed.

---

### Meta-Review · Area_Chair_PvV9 · 2023-09-18

**Recommendation:** 3

**Metareview:**

This paper presents a study comparing the performance of different combinations of domain-expert models against a big model trained on a combination of the data of all domains. All the reviewers agree that the work is sound. One common concern raised by the reviewers is that results are only presented on one language pair (en-ja), however the authors pointed out that according to the guidelines this should be enough for a a short paper and I agree with that. It presents some initial experiments that can trigger further research.

It is also worth noting that there was some confusion among the reviewers concerning the access to references. The authors addressed these issues in their response and I want to reiterate this here: MBR and QE do *not* have access to the references, thus they are fair methods to include in the comparison. Oracle does have access to the references, and therefore it is marked as such and provides an upper bound to the performance of the model. This does not compromise the soundness of the paper.

The weak point of this work in the excitement. None of the reviewers was overly enthusiastic, and I tend to agree with their judgement. There are no new methods presented in the paper, and it mainly consists of the experimental comparison between the presented ones. It can constitute a useful resource for practitioners, but I'm a bit reluctant to allocate a slot in the main conference for it. I think Findings would be a better destination, where it can constitute a reference to trigger further research.

---

### Decision · Program_Chairs · 2023-10-07

**Decision:**

Accept-Findings

**Comment:**

This paper presents a study comparing the performance of different combinations of domain-expert models against a big model trained on a combination of the data of all domains. All the reviewers agree that the work is sound. One common concern raised by the reviewers is that results are only presented on one language pair (en-ja), however the authors pointed out that according to the guidelines this should be enough for a a short paper and I agree with that. It presents some initial experiments that can trigger further research.

It is also worth noting that there was some confusion among the reviewers concerning the access to references. The authors addressed these issues in their response and I want to reiterate this here: MBR and QE do *not* have access to the references, thus they are fair methods to include in the comparison. Oracle does have access to the references, and therefore it is marked as such and provides an upper bound to the performance of the model. This does not compromise the soundness of the paper.

The weak point of this work in the excitement. None of the reviewers was overly enthusiastic, and I tend to agree with their judgement. There are no new methods presented in the paper, and it mainly consists of the experimental comparison between the presented ones. It can constitute a useful resource for practitioners, but I'm a bit reluctant to allocate a slot in the main conference for it. I think Findings would be a better destination, where it can constitute a reference to trigger further research.